# Immunomodulatory Strategies for Parapoxvirus: Current Status and Future Approaches for the Development of Vaccines against Orf Virus Infection

**DOI:** 10.3390/vaccines9111341

**Published:** 2021-11-17

**Authors:** Alhaji Modu Bukar, Faez Firdaus Abdullah Jesse, Che Azurahanim Che Abdullah, Mustapha M. Noordin, Zaharaddeen Lawan, Hassana Kyari Mangga, Krishnan Nair Balakrishnan, Mohd-Lila Mohd Azmi

**Affiliations:** 1Department of Pathology and Microbiology, Faculty of Veterinary Medicine, Universiti Putra Malaysia, Serdang 43400, Selangor, Malaysia; noordinmm@upm.edu.my (M.M.N.); gs53807@student.upm.edu.my (Z.L.); gs5186@studentupm.edu.my (H.K.M.); krishnan_ukm@yahoo.com (K.N.B.); 2Department of Science Laboratory Technology, School Agriculture and Applied Sciences, Ramat Polytechnic Maiduguri, Maiduguri 1070, Borno, Nigeria; 3Department of Veterinary Clinical Studies, Faculty of Veterinary Medicine, Universiti Putra Malaysia, Serdang 43400, Selangor, Malaysia; jesse@upm.edu.my; 4Department of Physics, Faculty of Science, Universiti Putra Malaysia, Serdang 43400, Selangor, Malaysia; azurahanim@upm.edu.my

**Keywords:** parapoxvirus, vaccine, immunomodulators, vaccination, Orf virus

## Abstract

Orf virus (ORFV), the prototype species of the parapoxvirus genus, is the causative agent of contagious ecthyma, an extremely devastating skin disease of sheep, goats, and humans that causes enormous economic losses in livestock production. ORFV is known for its ability to repeatedly infect both previously infected and vaccinated sheep due to several immunomodulatory genes encoded by the virus that temporarily suppress host immunity. Therefore, the development of novel, safe and effective vaccines against ORFV infection is an important priority. Although, the commercially licensed live-attenuated vaccines have provided partial protection against ORFV infections, the attenuated viruses have been associated with major safety concerns. In addition to safety issues, the persistent reinfection of vaccinated animals warrants the need to investigate several factors that may affect vaccine efficacy. Perhaps, the reason for the failure of the vaccine is due to the long-term adaptation of the virus in tissue culture. In recent years, the development of vaccines against ORFV infection has achieved great success due to technological advances in recombinant DNA technologies, which have opened a pathway for the development of vaccine candidates that elicit robust immunity. In this review, we present current knowledge on immune responses elicited by ORFV, with particular attention to the effects of the viral immunomodulators on the host immune system. We also discuss the implications of strain variation for the development of rational vaccines. Finally, the review will also aim to demonstrate future strategies for the development of safe and efficient vaccines against ORFV infections.

## 1. Introduction

Small ruminants are of great economic importance to humans as they provide a tremendous source of protein, calcium, vitamins, fiber, hides, and especially wool for hundreds of millions of people worldwide. However, it is believed that the economic losses caused by viruses are not adequately recorded because it is an acute infection. Moreover, a larger percentage of infectious diseases that hinder the livestock industry today are caused by pathogenic animal viruses [1,2,3,4,5]. To meet the high demand in livestock production, animals must be protected from infectious viruses such as ORFV [6,7,8,9] and several other pathogenic viruses of veterinary importance [10,11,12,13,14,15,16]. Therefore, the prevention of infectious diseases in animals can lead to widespread eradication of viruses [17,18,19,20], sustainable livestock production [21] and consequently improved life expectancy of farmers and veterinarians [6,22,23,24].

Orf virus (ORFV), the prototype species of the genus Parapoxvirus in the family Poxviridae causes highly contagious skin lesions popularly known as orf in several species of small ruminant particularly sheep and goats [17,25,26,27,28,29,30], although it can also infect humans [31,32,33,34]. ORFV infection may result in up to 100 percent morbidity in lambs, mortality can be up to 50 percent in animals with secondary infections [35] and immunocompromised conditions [36]. The virus is well known for its resistance to harsh environmental conditions [37] and its ability to cause persistent reinfections in sheep [38] and goats which has contributed to the further spread of infections in other animals such as deer, guinea pigs, and dogs [28,30,31,32,33,34,35,36,37,38,39]. The virus can be inadvertently transmitted to a susceptible animal through injured skin, readily replicates, and causes proliferative lesions that give rise to scab [19,20,21,22,23,24,25,26,28,29,40]. The classic pathological signs are scabs on the mucosal borders of the skin, lips, nose, eyelids, feet, teats of lactating ewes, and mouth [26,31,32]. Likewise, the virus incurred huge financial losses in the small ruminants industry by reducing both the quality [39,41] and quantity of milk [3,42,43], and even the death of infected animals [6,44,45]. In addition, ORFV infection has serious socio-cultural and economic challenges for livestock farmers, most of whom rely on small ruminants as their main source of livelihood [17,27,33,34,35,36,37]. Thus, ORFV infections increase the demand for the huge cost of management; resulting in reduced average income [27,40,46,47,48,49]. However, the acute infection caused by ORFV, which usually affects the lips, nose, teats, and sometimes on mucosal parts, has been described as a self-limiting disease, [32,50,51,52,53,54,55,56,57], and as such it has no positive impact on management although in several instances, particularly in lambs, it can turn to be persistent and even death [32,48]. The severe infection, which usually affects the eyelids, feet, vulva, and in some cases the udder has reportedly led to outbreaks in young animals [37] and kids resulted in tremendous losses in dairy farming across the world [20,47,49,50].

Furthermore, several strategies have been explored to develop effective antiviral drugs and prophylactic vaccines to control ORFV infection in sheep and goats, most of which are derived from immunogenic envelope proteins. Treatment of infected animals with antiviral drugs can only reduce the severity of the disease but cannot eliminate the virus [37,51]. Although, treating an infected animal with antibiotics can only minimize secondary bacterial infections such as staphylococcal dermatitis associated with ORFV infection [20,58,59,60,61,62,63,64,65]. It is well documented that misuse of antibiotics and chemotherapeutic agents has resulted in acaricide antibiotic resistance and accumulation of residues in the affected animals [41,52,53,54]. So far, vaccination has been the most effective alternative to antibiotics and antiviral agents [66]. Thus, the use of prophylactic vaccines and quarantine are the most cost-effective approaches to minimize the need for antimicrobials by controlling ORFV infections in sheep and goats of all ages [67,68,69,70]. As such, strict compliance to vaccination, extensive precautions combined with prompt treatment of infected animals, [71,72,73] and other containment measures such as improved sanitation appear to prevent further spread of the disease [74,75,76,77,78,79].

To date, vaccination of an infected animal is the best, if not the only, alternative, to effectively eradicate ORFV [78,79]. Vaccines are biological or synthetic preparations of immunogens-proteins, carbohydrates, and lipids derived from microorganisms or other synthetic sources- that when administered to the host, induce sustained immunogenic memory for the antigen [79]. Vaccines are part of an anti-infection medicine that promotes the stimulation of the desired immune response against infectious agents in both humans and animals of veterinary importance [79,80,81,82,83,84]. Vaccines intended for animals and humans should be safe, stable, highly immunogenic, and most importantly excellent at eliciting protective immunity [78,85]. Therefore, vaccines against ORFV should be used for recurrent infections and vaccinated sheep or goats should be kept isolated from unimmunized animals [48,85]. Nevertheless, lambs and young animals lacking ORFV-specific antibodies can acquire significant immunity if immunized within the first week after birth [86]. However, there are few scientific publications on the development of vaccine and vaccination strategies against ORFV infections [86,87,88]. Additionally, the existing live-attenuated vaccines against ORFV infections do not elicit protective immunity against the virus [56,88]. Therefore, current attenuated vaccines present a serious risk of reversion to virulence [56,89]. These varying degrees of vaccine failure warrant the re-examination of several determining factors particularly genes and proteins involved in virulence and pathogenesis of the virus, which may influence the safety and efficacy of existing vaccines against ORFV infection [53,60,90,91,92,93].

Detailed knowledge of viral structure and its immunomodulatory properties has the potential to reveal underlying factors that promote more rational vaccine design [56,88,89,94]. ORFV is an ovoid epitheliotropic linear double-stranded DNA [59,95] with dimensions of approximately 260 nm × 160 nm [9,96]. The genome is approximately 138 kbp long and encodes 132 genes [93,97]. The most important physical feature of this virus is the tubular, filamentous arrangement of the outer layer of the viral particle [97,98] as shown in Figure 1. The viral capsid proteins are arranged on the surface in tabular filamentous structures [99]. The inner membrane of the virion particle surrounded by a thick wall consists of the nuclear membrane, the palisade layer, and the nucleoprotein [63,100]. The 138 kbp ORFV genome consists of 17% highly variable genes involved in pathogenesis, virulence, host range, [101,102,103,104,105,106,107,108,109,110,111,112,113] and immunomodulatory activities of the virus [70,104]. The variable regions surrounded by 3 kbp of inverted terminal repeats (ITRs), which are similar but oppositely arranged nucleotide sequences, are located at the extreme terminal ends of the genomic DNA [104] and are strongly bonded with hairpin loops [32,93,105]. On the other hand, the central conserved genomic region of ORFV accounts for approximately 80% of the entire genome which is responsible for viral replication, transcription, and morphogenesis [32,54,63,105,106,107,108].

It is well-documented that major immunogenic proteins located within the conserved region particularly B2L (ORFV011) and F1L (ORFV059), are exploited prophylactically as vaccines against ORFV infections [78,109]. Stimulation of humoral and T-cell responses against such conserved (ORFV 011 and ORFV 059) antigens has been reported to elicit robust protection against ORFV infections [110]. Interestingly, the discovery of highly neutralizing antibodies and T-cells elicited by immunogenic proteins (ORFV 011 and ORFV 059) has led to the development of chimeric recombinant DNA and subunit vaccines [111]. A more recent study has demonstrated that the chimeric recombinant ORFV DNA vaccine has elicited vigorous neutralizing ORFV-specific IgG and T-cell (CD4+ and CD8+) responses in mice [109,110,111].

This review is aimed at a broader approach to vaccine development against ORFV infections. The review also discusses the impact of the host immune response against ORFV infections, paying special attention to the efficiency and efficiency of existing vaccines to provide in-depth knowledge into the major challenges faced by existing vaccines. In addition, the article also focuses on the impact of immunomodulatory proteins encoded by ORFV and the role of some highly conserved genes, encoding immuno-dominant proteins, to provide a new strategy for future vaccine development. The well-documented problems of genetic variation among ORFV strains will also be discussed. Likewise, the review will also seek to demonstrate new strategies and future approaches for the development of efficacious vaccines to control ORFV infections. Such challenges can be addressed through careful observation and the use of recent advances in molecular biology to fill the existing knowledge gaps.

## 2. Mechanisms of Immune Response to ORFV Infection

### 2.1. Overview of Current Knowledge in Immune Responses Elicited by ORFV Infections

The immune system is versatile, and the collective network of specialized cells, tissues, and organs has evolved to protect the host from invasion by microorganisms and their toxic products [68,112]. The immune response to ORFV infection consists of both innate and adaptive responses [9,113]. Despite several studies conducted on the immune response to ORFV infection, many mechanisms of host interaction with the virus are still not clearly understood [114,115,116].

However, ORFV penetrates a susceptible host and temporarily replicates transiently by undermining host immunity [117]. Once the viral pathogen is internalized, the cells of the innate immune system such as neutrophils, MHC class II-dendritic cells (DCs), and natural killer (NK) cells are constantly recruited to the sites of infection to capture the viral antigens, which facilitates the migration of the captured antigens to the peripheral lymph node where the antigen presentation to newly recruited naïve T-cells and memory cells take place [82,117,118,119]. The dendritic cells (DCs) facilitate the transport of the internalized virus to the lymph nodes for further proliferation [119] and the presentation of specialized cytokines such as granulocyte-macrophage colony-stimulating factor (GM-CSF), interleukin-1β (IL-1b), interferon-α (IFN-α), and IL-8, and IFN-γ [54,56,73,80,120]. In addition to innate immune cell recruitment, B-cell, CD4^+^, and CD8^+^ T-cell mediated immune responses are the most common adaptive immunity associated with infected host cells [121]. However, the activated T-cell immunity could not destroy this antigens because the virus releases several virulence factors that prevent possible neutralization of the virion particles, as shown in Figure 2 [122].

### 2.2. Innate and Adaptive Immune Responses against ORFV Infection

Innate immunity is the first form of defense mechanism against an invading virus and is considered a nonspecific response due to the lack of immunological memory [55,69,79,123]. In addition, ORFV has strong immunomodulatory activities [118] and has evolved a strategy to evade the immune system by developing a number of virulence factors [124]. The virus targets numerous non-specific mediator defenses including complement, chemokines interleukins, inflammation, interferons (IFN), and tumor necrosis factors [79,125]. Another important line defense next to the innate response is the adaptive defense mechanism, which includes antibody-mediated and cell-mediated responses [83,111,125].

Adaptive immunity is often referred to as cell-mediated and antibody-mediated responses that promote host cell recognition by invading viral antigens [65,70,126] and tailoring immune system adaptation to a specific antigen [127]. Several studies suggested that T-cells play an important role in eliciting protective immunity against ORFV [83,128]. However, the ability of the virus to repeatedly infect the susceptible host does not appear to be associated with the production of the desired memory T-cells as a promising hypersensitivity reaction to ORFV antigen, as reported by several recent publications [54,79,83,129]. Thus, such an immune response cannot be considered abnormal for ORFV infection, as significant levels of CD4^+^ T-cells have been detected among the infected animals when compared to CD8^+^ cytotoxic T-cells and/or B-cell response [115,130,131,132,133,134,135]. On the other hand, the humoral immune response does not play an important role in protecting animals from ORFV infection [136]. It is well-documented that the abundant antibody titers detected in the sera of infected sheep with ORFV, the passage of antibodies from sheep to their offspring via colostrum [115,137] or, inoculation of sera from infected animals to healthy animals do not elicit protective immunity [138]. Therefore, immunity against ORFV is not long-lived and the virus can readily reinfect both vaccinated and unvaccinated sheep [38,136,139]. Interestingly, smaller lesions that resolve more rapidly have been observed in the case of vaccinated and reinfected sheep [67,136,140]. Presumably, the absence of ORFV-specific neutralizing antibodies in the vaccinated or previously infected animals is the main reason why ORFV to causes repeated reinfections in its host [136,141]. In addition to the lack of neutralizing antibodies, the ability of ORFV to repeatedly reinfect previously infected and/or vaccinated sheep is attributed to the immunomodulatory genes and proteins encoded by the virus that temporarily suppress host immunity [142].

### 2.3. Immunomodulatory Properties of Orf Virus

Parapoxvirus (ORFV) is well recognized for its ability to withstand numerous challenges to the host-triggered immune response. ORFV is able to partially circumvent the host immune response because upon acute infection of the mucocutaneous boundaries of the epidermis, the virus immediately releases multiple virulence proteins that subvert host immunity [122,143]. Therefore, the virus may ultimately utilize the various immunomodulatory strategies to modulate, subvert, evade, [108] and/or suppress host immunity [144,145,146]. The major immunomodulatory genes (IMGs) encoded by ORFV include chemokine binding protein (CBP), interleukin-10 (IL-10), vascular endothelial growth factor (VEGF), the GM-CSF inhibitory factor (GIF) [98,105] and interferon-resistance gene (OVIFNR), which inhibits protein synthesis by preventing an enzyme dsRNA-dependent kinase [48,68] (Table 1). It is well-documented that IMGs and immunomodulatory proteins (IMPs) of ORFV are acquired either from vaccinia viruses (VACV) [119,147] as a result of continuous interactions with their host [44,91]. For instance, CBP, GIP, VIR, and dUTP were acquired from VACV homologue proteins [44,105,148] and their origin from the ancestral poxviral genes [92,93,94]. Thus, the discovery of immunomodulatory genes [98] and proteins secreted by ORFV may explain how the virus escape elimination by host immunity [105,149].

In addition to therapeutic benefits, IMPs are highly involved in viral pathogenesis, and virulence activities [88,105,149]. Accordingly, such advantages would draw attention to the immunology [45,91,150] and the pathogenesis of the virus [53,85], as well as potential treatment opportunities [88,96,151]. On the other hand, intracellular virus also produces immune-modulatory proteins (IMPs) that subvert the host immune response, further blocking cell interactions [45,152].

One of the well-characterized immunomodulatory genes is the ORFV orthologue of interleukin-10. ORFV-IL-10 (ORFV127) is one of the early genes [99,153] synthesized by the virus and is located at the terminal right end of the viral genomic DNA and encodes a ~550bp, 21.7kDa protein [97,106,111,113,154]. Interestingly, parapoxvirus (ORFV) is not the only virus in the family of poxvirus that can secrete the vIL-10 homologue, which is actively involved in the suppressing and subverting of the host cellular [106] and humoral immune responses [85,103,155]. In addition, ORFV-IL-10 protein can inhibit the synthesis and trafficking of host cytokines and chemokines [103,106,156] such as IL-1b, IL-8, TNF-a, IFN-γ which is likely inhibit inflammatory reactions that in turn can prevent recruitment [157] and activation of immune cells at the site of the infection [99,108,109]. In addition, ORFV-IL-10 induces the proliferation of thymocytes and mast cells [72]. Nevertheless, ORFV-IL-10 also modulates and/or inhibits T-cell proliferation by suppressing the functions of MHC II (major histocompatibility complex class II) molecules, which decrease the recruitment of innate immune cells such as mast cells, macrophages, monocyte, and dendritic cells (DCs) to the sites of damaged skin [100,110,158].

Another virulence factor of parapoxvirus, ORFV that suppresses host immunity is chemokine-binding protein (CBP), which is encoded by the gene ORFV112 (~861 bp) [99,103,104,105,158]. ORFV-CBP is one of the early virulence genes synthesized by the virus after successful host cell invasion [104,159]. CBP is located in the highly variable terminal regions of the Orf viral genome and encodes a 31.18kDa protein [104,157]. It is well documented that the ORFV-CBP (ORFV112) allows the temporary replication of viral antigen in infected cells by inhibiting the reactivation of cellular immune responses such as IL-1β, IFN-α, IL-8, and IFN-γ [80,157]. The ORFV-CBP protein inhibits the recruitment and migration of dendritic cells and other immune cells to peripheral lymph nodes to activate an adaptive immune response [87,157]. Unlike other immunomodulators, CBP has no mammalian homolog [105]. In addition, the lack of trafficking of the immune cells can lead to inhibition of the MHC class-II pathway [102,106,157], affecting the recruitment and/or activation of cytotoxic T cells (CD8**^+^**T-cell) at the site of infected skin [87,106,107,160].

ORFV also encodes a vascular endothelial growth factor (VEGF) gene, one of the immunomodulatory proteins responsible for inhibiting the proliferation of host immunity [94,160]. However, host VEGF is a regulatory protein actively involved in the elimination of tumors [112], virus-infected cells [113], and in the wound healing process [114]. The genetic analysis of ORFV has shown that the ORFV-IL-10 [106,160] and the ORFV-VEGF (ORFV132) genes were purloined from host organisms during the evolutionary coexistence [85,115]. Therefore, the viral protein ORFV-VEGF is characteristic only of viruses of the genus Parapoxvirus. ORFV132 (~462 bp) is also one of the early genes of the virus and is located in the highly variable terminal regions at the right end of the conserved region [84,88,94,97,116]. Previous studies indicate that the ORFV-VEGF may promote the continued proliferation of epithelial cells which would facilitate the creation of more binding sites for ORFV replication [94,116,160]. In addition to the viral replication, ORFV-VEGF protects the virus from the violent effects of collective immune responses and subverts the effects of host antiviral apoptosis [55,69]. This suggests that ORFV-VEGF enhances intracellular replication of the virus by subverting the apoptotic activity of the host cell [92,98]. However, recent studies have shown that knockout of the ORFV-VEGF protein from genomic DNA attenuates has been reported to attenuate the virus and presumably reduces the severity of the disease in the host cells [84,94,160].

On the other hand, GIF (GM-CSF inhibitory factor) is a hemopoietin produced by macrophages and T-cells (among other cell types) that stimulates the development of neutrophils, macrophages/monocytes, and eosinophils from hematopoietic progenitor cells [88,103,110,160]. Unlike the other immunomodulatory proteins, ORFV-GIP has no counterpart in the host genome. Interestingly, a more recent study revealed that ORFV-GIP is one of the new immunomodulatory proteins that suppress the functions of the host cytokines such as IL-2 and GM-CSF [65]. In addition, the interferon-resistance genes of ORFV act within an infected host cell to inhibit the possible anti-inflammatory proteins blockage induced by interferons, on the ORFV orthologue of interleukin-10 and ORFV GM-CSF inhibitory factor (GIF) proteins are released from infected tissues and induce their immunomodulatory activities away from infected cells [103,157,158,159,160]. Therefore, blockage of the activities of both the innate and acquired immune response is one of the main strategies in successful ORFV infections. However, the functional mechanisms of GIP in ORFV pathogenesis and virulence are yet to be determined [84,117,160].

## 3. Overview of Vaccines against Animal Viral Infections

### 3.1. Common Viral Vaccines of Veterinary Importance

Prophylactic animal vaccines have played a significant role in the eradication of contagious diseases [5,14,25,160]. So far, vaccination has been one of the efficient and cost-effective means of preventing viral infections of both humans and veterinary importance [157,159,160,161]. Besides the intended improvement of animal health and productivity, veterinary vaccines play a tremendous role in safeguarding consumers’ health by offsetting the use of antimicrobials and antiviral drugs for the treatment of animal diseases [56,161]. Interestingly, the livestock sector has grown consistently as the result of proper and timely animal vaccinations with the existing vaccine platforms shown in Table 2 [84,162]. There are six main types of viral vaccines that are commonly used in animals; live attenuated, DNA vaccines, inactivated vaccines, recombinant vaccines, subunit vaccines, and peptide vaccines [8,120,162].

### 3.2. Live Attenuated Vaccines

Despite their drawbacks, live-attenuated vaccines have been the most effective and cost-effective strategy for eradicating ORFV [74,162]. The virus possesses a peculiar capacity to modulate and hence suppresses host immunity, which warrants further improvement [74,162]. The live-attenuated vaccines, commonly known as live attenuated vaccines are highly effective in eliciting long-lived immunity against infectious organisms. The main characteristics of live-attenuated vaccines are the elimination or complete loss of virulence factors while retaining the immunogenicity of the pathogen [144,162]. The live-attenuated vaccines can be produced by continuous passages of the virus in vitro cell culture-based [56] or in vivo chicken embryonated eggs [78,162], with the prime target of losing the virulence factors, but still retaining immunogenicity pathogens [163]. The production of cell-culture-based, and egg-based attenuated vaccines have been in use for more than two decades, many are being licensed for veterinary use [120,162]. So far, attenuated vaccines have been developed for numerous animals including cats, horses, dogs, cattle, and other domesticated animals worldwide [103,163]. This vaccination strategy offers some considerable advantages, not the least of which is a desirable immune response against viral diseases that can induce long-lived immunity [124]. Attenuated vaccines can usually elicit durable immunity after a single injection and additional boosters aren’t required [130]. The ORFV D1701 strain is an excellent example of an efficient attenuated vaccine against ORFV infection in sheep that requires a single or two boosters to induce durable immunity [128,163]. Despite these advantages, several drawbacks have been reported ranging from reversion to virulence [25], gene deletions, mutations, and vaccination failure have been observed so far as the results of the replication nature of the vaccine strains that would enable the virus to multiply in the vaccinated animals [8,120,162,163] (Table 2). Furthermore, vaccines produced by live virus modification are reported to have difficulty in understanding the true nature of the immune response as well as difficulties in optimization during administration and subsequent boosters [116,129]. In addition, the conventional viral attenuation approach is highly time-consuming and not promising against highly virulent ORFV variants.

### 3.3. Inactivated (Killed) Vaccines

Inactivated vaccines are prepared by killing or inactivating the viral antigens by heat or chemicals such as formaldehyde and/or formal saline [25]. Thus, inactivated vaccine is relatively safe, cannot reverse virulence, and most importantly, it can be used to vaccinate even immunocompromised animals [121,162,163,164,165]. It is well-documented that inactivation facilitates the induction of Th1and other T-cell mediated immune responses. Despite these benefits, inactivated vaccines can induce a short-lived immune response [144,165]. As such, there is a need for adjuvants in the vaccine formulations to improve immunogenicity [84,166]. In addition, inactivated vaccines are relatively expensive to produce due to the additional cost involvement of an adjuvant, purifications, chemicals for inactivation, and most importantly cost implications of multiple uses of booster dosage [121,167]. The inactivated or killed vaccines are further classified into several types including; (1) Inactivated complete (whole) viral pathogens by heat or chemicals e.g., scab-based inactivated vaccine against Orf virus and rabies vaccine; (2) purified protein-based vaccine obtained from killed viral particles or purified antigens produced [120,168].

### 3.4. Subunit Vaccines

Subunit vaccines are a sub-type of killed vaccines that contain purified or recombinant fragments obtained from the desired genes that are cloned and expressed in a bacterial vector [120,163,164,165,166,167,168]. Therefore, the production of subunit vaccines often involves the ultimate use of genetic engineering techniques that target specific immunogenic proteins of a particular virus that can trigger the production of durable humoral and cellular immunity [168,169]. The production of subunit vaccines requires: (1) careful selection of immunogenic microbial protein, (2) efficient adjuvant and (3) suitable delivery systems [164]. In comparison to their live-attenuated equivalents, subunit vaccines have shown numerous advantages: minimal or no adverse effects, less reactogenic profiles, chemically defined, inexpensive, stable shelf life, and most importantly can be used for vaccination of immunocompromised animals [169]. However, the downside of this type of vaccine may confer lower immunity, and therefore, there is a need for multiple booster doses vaccination [170]. It has been reported that rabbits vaccinated with highly immunogenic F1L subunit recombinant proteins derived from ORFV induce the production of humoral immunity [78,109,170].

### 3.5. Recombinant Live Viral Vaccines

Recombinant live vaccines are viral vector-based immunogenic products that are made of a live virus strain as a vector carrying an encoding target gene interest [157,159,160,161,162]. Thus, large DNA viruses such as poxviruses, herpesviruses, and adenoviruses are commonly exploited as helper-independent vectors for the expression of an immunogenic gene of the desired virus [124,169,170]. Recombinant DNA technology has made it possible to use viruses as vectors for the expression of a wide variety of genres. Therefore, it is scientifically applicable to use existing live-attenuated vaccines such as the one derived from ORFV as a destination vector to express certain immunogenic proteins of the viral pathogen as vaccines [74,103]. Interestingly, this class of vaccine is gaining greater attention due to its ability to induce neutralizing antibodies and cytotoxic T-cell immune responses [124,170]. A recent study demonstrated that the product of recombinant ORFVD1701-V-RabG carrying the rabies viral protein has elicited a durable immune response after a single dosage of dog vaccination [157,159,160,161,162,163,164,165,166,167,168,169,170].

### 3.6. DNA Vaccines

DNA vaccines, less commonly-known as nucleic acid vaccines, show significant efficacy owing to their huge capacity for conveying additional copies of genes that can lead to the production of chimeric (multivalent) vaccines [111,170]. The replication of parapoxvirus DNA vaccines in the host cytoplasm have shown to induce long-lived immune response at single time point injection [120,150,171]. Thus, the advantages of this type of vaccine over the killed vaccines include prompt antigen presentation [25] and activation of the immune cells, relatively lower cost of production, and highly efficacious in small ruminants [86,121,122]. DNA vaccines are emerging as novel and essential strategies for animal vaccine development [4,12,52,64,170]. Such a type of vaccine is developed from a plasmid encoding a particular gene of interest that is derived from a target viral pathogen. Therefore, the gene of interest can be cloned into designated expression vectors, together with suitable genetic elements [122,171]. The direct injection of such plasmid encoding the gene of interest confers sustained immune responses [172,173]. Despite the advantages, there are serious health concerns to both animals and meat consumers owing to the presence of an antibiotic-resistant gene contained in the vector backbones [8]. Another drawback of this class of vaccine is its limitation to small animals such as sheep, goats, cats, and fish [122,173]. However, DNA vaccines against ORFV infection have been reported to trigger T-cells and ORFV-specific IgG in young and adult mice [124,173].

**Table 2 vaccines-09-01341-t002:** Common viral vaccines of veterinary importance.

Vaccine Type	Advantages	Disadvantages	Beample(s)	Immune Response	Ref.
Live attenuated vaccines	They are highly immunogenic and are excellent for inducing sustained cell-mediated and humoral immunity with a single dose.	Risk of mutation and reversion to virulent strain in an immunised host. High cost of production and storage facilities.	Parvovirus, adenovirus-2 vaccines Rubella (MMR), Influenza,	Evoke vigorous T- and B-cell responses.	[134,157,169,170]
Inactivated vaccines	They are highly safe and are made from non-replicating whole-cell viruses that pose no risk of reversion to cause disease.	Poorly immunogenic and inefficient to stimulate prolonged duration of immunity, so vaccination of 2 to 4 weeks is needed	Rabies Vaccines Feline Leukemia vaccines, canine influenza vaccines	Immune response T helper 1-type cytokines	[151,168]
Subunit vaccines	Poorly immunogenic and inefficient to stimulate prolonged duration of immunity, so vaccination of 2 to 4 weeks is needed	Weaker immune response and it requires adjuvants	F1L recombinant protein was able to stimulate the production of antibody	ORFV-specific IgG,	[25,84,124,170]
DNA vaccines	induces both cellular and humoral immunity, ensuring a sustained immune response once the animal encounters the wild-type virus at a later time	Lower efficacy in large animals and serious safety issues due to activation of oncogenes and antibiotic resistance.Virus-specific IgG, T-cell immune responses.	Influenza and Herpes vaccines, parapoxvirus vaccines	Viral specific IgG, T-cell immune responses.	[8,134,150]
Peptide Vaccines	effective in preventing animal infections no possibility of return to pathogenic phenotype.	controlled antigen displays and relatively low immunogenicity.	Influenza vaccines	Evoke CD4^+^, CD8^+^, B-cells and IFN-γ	[165,166]
Recombinant Vaccines	Attenuated virus used to introduce microbial DNA into host cells. It induces a strong immune response.	Risk of mutation and reversion to a virulent strain in an immunized host.	ORFVD1701-V-RabG Recombinant vaccine	Elici tviral-specific IgG, and T-cell	[3,25,86,160,168]

## 4. Current Status of Vaccines Development against ORFV Infections

### 4.1. The Current State of ORFV Vaccines

The historical events in vaccine developments didn’t just begin with the initial inoculation of human skin with the material collected from cowpox pustules by Edward Jenner to protect humans against smallpox [120,121,122,123,124,125,126,127,128,129,172,173]. Rather, it can be traced back to the history of infectious diseases of the human and veterinary importance [129]. Historically, ORFV vaccines containing tissue culture attenuated live virus or live scab-based virus preparations have been available since the late 1930s [25,86,129,159]. Parapoxvirus (ORFV) is currently used as attenuated live virus and/or as a vector for antigen delivery systems for other viral vaccines such as vectored rabies vaccines [25,78,128,174]. In spite of the abundant studies published, on both vaccine development ‘and immune response to ORFV infections, few vaccines against ORFV infection are licensed [56,174]. As of today, there are no universally approved sheep or goat vaccines against ORFV infections. Therefore, there is a need to develop a universal ORFV vaccine that could provide protection against worldwide strains of the virus [78,122,174]. However, the humoral, and T-cell analysis revealed that live-attenuated vaccines have shown several benefits. In addition, all the current licensed vaccines used to protect sheep and goats against ORFV infections are based upon live (scab-based) and live-attenuated (Figure 3). Currently, no subunit, peptides, or DNA vaccines are licensed, but some recent publications have demonstrated that chimeric DNA vaccines and recombinant vaccines against ORFV infection have shown promising results [126,129,175].

### 4.2. Safety and Efficiency Profile of Current Vaccines against ORFV Infections

This section discusses currently available vaccines against ORFV infections to analyze the safety, efficiency, effectiveness, and limitations of the existing vaccine platforms [144,174]. Vaccines are the most powerful approach to prevent infectious disease outbreaks [141,176]. So far, vaccination has been one of the most effective strategies designed to prevent ORFV infections [128,174]. The commercially available vaccines for vaccinating goats and sheep against ORFV infections are generally classified into two types: purified scab-based vaccine [176,177] and cell culture-based live-attenuated vaccine. The first cell culture-based vaccine against ORFV infection was developed in the 1930s at Texas Agricultural Experiment Station in Sonora, Texas (USA) [56,120,124,176]. After the successful discovery of the first vaccine, several other researchers made it their mission to improve the live-attenuated vaccines [78,120,176]. However, the live attenuated vaccine has been reported to provide short-lived protection against ORFV infection of up to 4–6 months [78,178].

The scab-based vaccine was prepared from a virulent ORFV strain derived from goat or sheep scab mouth virus [5,14,150]. However, live scab-based vaccines can induce a robust immune response [179]. Unfortunately, the development of such a vaccine is labor intensive [56,86,179] and it can serve as a potential source of environmental contamination by scab derived from vaccinated lesions [180]. Therefore, cell culture-based vaccines are relatively more preferred owing to increased safety as compared to the counterpart of scab-based vaccines [56,120,178]. Several studies suggest that the vaccine developed based on tissue culture has significantly higher efficacy [120,144,181] and safety compared to the scab-based vaccine [182]. 

At present, live-attenuated vaccines derived from the ORFV field strain are used prophylactically to prevent ORFV infection in sheep and goats, which play a significant role in reducing disease outbreak [182]. Unfortunately, the current vaccine platform is not safe or able to induce protective immunity and the virus can repeatedly reinfect the vaccinated animals [129]. For example, despite the immune response elicited by the current attenuated vaccines, ORFV continues to infect the susceptible animals as the results of the inefficiency of the vaccines [5,14,150,182].

In addition to the short-lived immune response, live-attenuated vaccines also present a serious risk of reversion to virulence [128,129,179]. The reason for reversion to its original virulent strain is still not clear but could occur due to possible deletion of one or more terminal genes encoding immunomodulatory proteins of the virus leading to attenuation of ORFV in sheep and goats as the results of long-term adaptation of the virus in tissue culture could render current vaccines inefficient [124,125,126,127,128,129,130,131,132,133,134,135,136,137,138,139,140,141,142,143,144,145,146,147,148,149,150,151,152,153,154,155,156,157,158,159,160,161,162,163,164,165,166,167,168,169,170,171,173,174,175,176,177,183]. Other reasons for vaccine failure include possible contamination of the environment by scab derived from vaccinated animals [178], improper storage of the vaccine, genetic [78,84,182] and physiological variation between animals, [121,144,182] and antigenic variations between the vaccine and field strain with increased virulence [55,56,57,61,62], nutritional factors, physical factors [59], and coinfection by bacteria [58,59,75,139,140,141,142,182]. Such factors may influence one or more features for the development of an effective vaccine, which may enhance the desired immune response against the target antigens [176,182]. Perhaps, the notable vaccine failure can be associated with one of these factors [129,144]. The effectiveness of current vaccine platform strategies against ORFV infections are shown in Table 3

However, the antibodies detected in the vaccinated animals are not considered significant for sustained protection against ORFV infection, but the presence of an antibody in the vaccinated sheep indicated prior exposure to ORFV [124,129,179,180,182]. On the other hand, live-attenuated ORFV vaccines based on cell culture elicited the desired immunity against originally virulent ORFV strain obtained from a British isolate of Orf virus [56,92,129,181]. On a contrary note, a tissue culture-prepared live attenuated vaccine derived from the OKA ORFV strain failed to protect sheep from ORFV infection because the vaccine did not induce a neutralizing ORFV-specific antibody [125] and/or T-cell immune response in sheep [14,56,78,124,159,182]. A similar study conducted at Western Texas demonstrated that ORFV the ORFV vaccine has been reported to cause an outbreak in vaccinated sheep and goats [146,179,180].

More recent studies demonstrated that plasmid DNA-based ORFV vaccines derived from virulent strains induced protective T cells and ORFV-specific antibody responses in neonates [3,27,64,130,135]. The result indicated that the plasmid DNA-based vaccines induced similar responses in mice compared to live-attenuated vaccines [179]. Zhao et al. also reported that DNA vaccines elicited ORFV-specific IgG and T-cell (CD4^+^ and CD8^+^) responses in mice after two repeated booster doses [56,132,133,134,135,136]. In another study, subunit ORFV vaccines derived from sheep scabs have been reported to induce an anamnestic response, with a significant increase in T-cell response and ORFV-specific IgG response against virulent strains [63,66,80,124,125,126,127,128,129,130,131,132,133,134,135,136,137,138,139,140,141,142,143,144,145,182].

### 4.3. Implications of Strain Genetic Variation for the Rational Design of ORFV Vaccines

A proper understanding of the implications associated with strain genetic variation among viral pathogens could pave the way for the design of universal vaccines with improved efficacy [60,76,144,168,169,170,171,173,174,175,176,177,178]. However, careful design of an efficacious vaccine candidate against viral infections is a huge challenge guided by years of studies on viral biology and host immune response [176,182,184]. Thus, strain genetic variation is an important characteristic of ORFV [171,184] and remains one of the major constraints of current vaccines against ORFV infections [178,179,180,181,182,184]. It has been observed that genetic variation can occur even within genes at the conserved regions leading to the alteration of structural proteins targeted in a possible universal vaccine formulation [92,145,171,173,174,175,176,177,178,179,180,181,182,184]. In addition to alteration in immunogenic proteins, ORFV can undergo highly antigenic variation in other essential proteins, which can often lead to continuous reinfection of the virus in previously vaccinated animals [125,145,157,177]. It has been reported that the possible outbreaks in vaccinated sheep often due to genetic reassortment between ORFV from district species [75,182]. The most obvious of genetic reassortment is that involving the viral defensive immunomodulatory proteins [180,182] or inverted terminal repeats (ITR) of the virus [171,174,182]. Therefore, genes in the terminal and ITR regions are highly variable [174] and responsible for antigenic variations, pathogenesis, virulence [184], and/or tissue tropisms [101,102,103,104,105,106,142,184]. It is well documented that knockout of an immunomodulatory protein (IMPs) conserved among parapoxviruses of the family poxviridae can reflect variability in animals [174] and tissue tropism facilitating the phenotypic variation observed at the point of ORFV disease manifestations [92,133,184].

Work by Cottone et al. reported that a highly attenuated tissue culture-based vaccine derived from ORFV strain D1701 stimulates an immune response against the wild-type ORFV infection [129,133,159]. Unfortunately, the immunity elicited has only lasted for about 4–6 months [3,60]. Thus, the immunity induced by the attenuated D1701 vaccine is inefficient in protecting the animal from reinfection [60,174,175,176,177,178,179,180,181]. Cottone et al. also reported that the enlargement of inverted terminal repeats (ITR) up to 18 kbp due to recombination between nonhomologous sequences during cell culture adaptation [133]. Immuno-modulation of the host defense mechanism is a common strategy used by the ORFV to overcome host-specific antibodies [133]. Thus, such variations at the genetic level may significantly contribute to poor immune response to vaccines, [177,178] and consequently, such critical insights could be useful strategies in future vaccine design and development [167,168,169,170,171,173].

Variation may occur in several ways depending on the biology of the causative agents [171]. For example, some animal viruses such as aphthoviruses, rhinoviruses, and parapoxviruses have been shown to have multiple serotypes circulating within a particular geographical region [25,78,144,160]. However, the rate of genetic recombination, as well as the emergence of new antigenic variants have not been clearly elucidated [63,66,80]. Genetic variations in the ORFV strain may be caused by several factors arising from a different mechanism, including: (1) genomic rearrangement and subsequent deletion of one or more genes as the result of the attenuation of the virus in tissue culture [58], (2) gene enlargement as a result of duplications in the ITR region, which can result in a huge increase in the overall genomic sequence of the virus, and (3) nucleotide substitution within the coding regions of GC-rich sequence [171,174,185]. Host variation can often lead to vaccine failure which is not insignificantly related to the fact that ORFV strains from sheep and goats are clusters but belong to different branches of the phylogenetic tree [59,63].

ORFV can also exhibit huge variability in circulating strains worldwide. It has been demonstrated that vaccination of goats with goat derived attenuated ORFV vaccine did not clearly minimize the severity of the lesions when it was exposed to field strain ORFV with increased virulence [78,120,124,173]. Additionally, vaccination of goats with the attenuated vaccine produced from relatively increased virulent field ORFV strains significantly reduced the number of previously infected animals but failed to reduce the severity of the infection [74,173,182]. This suggests that the vaccine does not provide cross-protection between the vaccine strains and field strains with increased virulence [179]. Recent studies demonstrated that the goats vaccinated with live-attenuated vaccines did not provide complete protection against ORFV infection when exposed to another ORFV strain derived from goats [124]. Such vaccination failure may have been related to the genetic factors that result in a serious outbreak of the disease in vaccinated animals [179]. In another study, sheep immunized with the live-attenuated vaccine developed from a virulent strain of ORFV did not elicit desired antibody and cell-mediated immune responses that can be seen with natural infections with field strain ORFV [79,144,157,159,160,161,162,163,164,165,166]. In several instances, lambs vaccinated with the scab-based vaccine developed from goats-based Orf virus failed to protect the animals from the heterologous virulent field virus strains [139]. Repeated re-infections [120]. and vaccine failures have been observed in both vaccinated goats and sheep [175]. Thus, such variations at the genetic level may significantly contribute to a poor immune response to vaccines, [177] and consequently, such critical insights could be useful strategies in future vaccine design and development [173,174].

### 4.4. Enhancement of the Current ORFV Vaccines

The tissue-culture-based live attenuated virus vaccines have played a significant role in preventing the spread of viruses [124,173]. Although live attenuated and scab-based vaccines protect animals ORFV infection worldwide, the continued reinfection vaccinated animals are a major concern that needs to be addressed [174]. Additionally, ORFV reinfection in vaccinated animals might be due to the global movement of vaccinated sheep and/or goats and genetic variation [171,173,174,175,177]. The enhancement of existing vaccines in terms of their safety, efficiency, efficacy, and cost-effectiveness is a key priority. Improvement of current vaccines can be achieved through careful selection of the desired ORFV strain for vaccine seed virus, proper use of tissue culture for vaccine seed virus propagation, and proper use of potent adjuvants and/or sustained delivery systems [152,153,154,155,168,169,170,171,173].

Over the last few years, several strategies have been explored to accomplish attenuation of the virus through genetic engineering rather than the traditional approaches; based on long-term adaptation of the virus in tissue culture [56,173]. In addition, advances in recombinant DNA technology and genetic engineering have opened a new avenue to enhance the current vaccine by deleting immunomodulatory genes of ORFV, to promote the development of vaccines with robust immune response [65,92,93,94,171]. A recent study indicates that the deletion of IMGs such as ORFV020, ORFV117 and ORFV132 from the Orf virus genomic DNA led to production of modified live-attenuated virus that can elicit protective immunity [92,98]. In addition, production of modified live-vaccine makes the virus a suitable vectored vaccine candidate [65,171,173,174,175,176,177,178,184]. Several studies demonstrated that the attenuation of vaccine strains by deletion of immunomodulatory genes has resulted in a significant increase in the immunogenicity of the vaccine [99,108,109,171]. Thus, the future vaccine can be engineered by targeting ORFV immunomodulatory proteins that inhibit the host immune response [85,92,93,94].

## 5. Future Strategies for the Development of Vaccines against ORFV Infection

In recent years, great success has been recorded in the development of vaccines. Recent advances in recombinant DNA technologies will continue to play a key role in shaping future ORFV vaccine development. Future vaccine design technologies would focus on various strategies that are particularly promising in their potential to mimic desired immune responses to challenging antigens for which current vaccination strategies have proven ineffective [167,174,175,176,177,178,179,180,181,182,184]. As with current ORFV vaccine platforms, there is a need for the development of a novel vaccine with increased efficiency/safety profiles [144,159,173].and improved adjuvants and sustained delivery strategies to overcome the current limitation observed in vaccines against ORFV infection [162]. The novel technologies are expected to improve the safety [144] and efficacy of future vaccines [144,174,182].

Furthermore, vaccines against ORFV and other small ruminants in the live-stock production industries are under huge demand in order to increase their application in the field. The future vaccine platforms should be designed to fulfill three main constraints; (1) compatibility with mass vaccination, (2) cost-effectiveness for large-scale production and delivery, (3) most importantly, they should be safe, effective, and efficacious. However, such constraints can serve as a benchmark for determining the efficacy of future vaccine platforms [4,25,84,173,185]. A schematic outline of some future vaccines against ORFV (Figure 4). The future vaccines against ORFV infections include nanoparticle-based subunit and/or peptide vaccines, DNA vaccine, and parapoxvirus vectored vaccines. Thus, the application of such vaccines can minimize the use of the attenuated virus that might easily reverse to virulent [147,174,184].

Subunit and peptide vaccines are composed of either glycoproteins [163] or proteins derived from specific ORFV genes that are actively involved in eliciting robust immunity [25,84,124,170]. Additionally, subunit and peptide vaccines are relatively safe [109,110,111] and easy to produce compared to efficient conventional live-attenuated and/or inactivated vaccines [170,182]. Unfortunately, subunit vaccines induce short-lived immune responses, so adjuvants are needed to boost immunity [152,153,154,155]. Recent significant advances in the field of nanotechnology has revealed that nanoparticles (NPs) are capable of enhancing the cellular immune response as well as the delivery of specific viral antigens to a target cell [154]. Therefore, nano-adjuvanted vaccines can be used to enhance the immunogenicity of existing and future vaccine candidates [168,169,170,171]. For example, peptide, subunit, and inactivated vaccine platforms are relatively safer but tend to be less immunogenic compared to live-attenuated viral vaccines [170]. Therefore, the low immunogenicity of such vaccine candidates can be enhanced by the addition of potent such as alum or nano-adjuvants that stimulate durable immune response [148,149,150,151]. Therefore, nano-adjuvants are used for vaccine development due following benefits: (1) enhancement of vaccine antigens efficacy; (2) entrapment and release of the vaccine antigens to the target cells in a sustained manner; (3) protection of delivered vaccine antigens from possible degradation; (4) it minimizes the dose, amount of vaccine antigen required vaccination in order to minimize production cost; (5) enchantment of humoral and cell-mediated immune response; (6) enhancing antibody specificity, affinity, avidity, and (7) prompt dissemination of the antigens to target cells [152,153,154,155,156,157,158,159,160,161,162,163,164,165,166,167,168].

Furthermore, due to the nanosized structure and shape, nanoparticles can be easily engulfed by the antigen-presenting cells, particularly the dendritic cells [144,150], and consequently trapped into the draining lymph nodes to stimulate the production of humoral and cell-mediated immunity [155,156]. Hence, the use of nano-adjuvants will further improve both the efficacy and effectiveness of vaccines [159]. If the concept of nanoparticle-based vaccines truly works, it would be a great approach in future strategies to control ORFV infections [158,159]. The immunity to nanoparticle-based (NPs) vaccines that result in enhancement of vaccine-stimulated immunity is summarized in Figure 5.

Peptide vaccines are a sub-type of subunit vaccine that are synthesized in vitro using peptides that containing immunogenic fragments critical immunodominant proteins of the target virus [157,159,160,161,162,163]. However, peptide-based vaccines are relatively easier to produce and show high stability than subunit vaccines (whole proteins). Interestingly, peptide vaccines have shown numerous advantages compared to their subunit counterpart, lower antigen complexity, lower toxicity, and most importantly low production costs [166]. Peptide-based viral vaccines can enhance immunogenicity by the addition of potent immunoadjuvants that stimulate the T-cell response (CD4^+^ and CD8^+^), sometimes the humoral response, and Interferon-gamma (IFN-γ) to destroy the virus from an infected cell. Currently, there are no peptide vaccines against ORFV and other viral infections. However, peptide-based vaccines against influenza virus and human papillomavirus vaccines have been reported to trigger the induction of desirable immune responses [167]. Thus, this class of vaccines can be developed using bioinformatics and molecular biological approaches to be designed in order to overcome the undesirable effects of conventional vaccines [164,182]. Interestingly, peptide vaccines are theoretically considered as a promising class of future vaccines due to the following advantages; easy synthesis, stability, and highly safe to be administered to a wider host range [165,166].

Recombinant viral vector vaccines are composed of a highly competent engineered vector backbone for the production of recombinant vaccine antigens [160,161]. The selected DNA fragment is subsequently recombined with a suitable expression vector that can lead to the production of the recombinant protein of interest [167,173]. Interestingly, advances in the understanding of host immunity to viral infections and pathogenicity coupled with efficient genetic engineering approaches have led to the development of novel vaccines [147,148,162]. Commonly used viral vectors include adenoviruses, alphaviruses, flaviviruses, herpesviruses, Newcastle disease virus, parvoviruses, and poxviruses have been manipulated to develop recombinant viral-vectored vaccines [160]. The most striking features of ORFV vectored vaccines include the induction of durable T-cells and humoral immune response, lack of systemic dissemination of the virus, and restricted target host. Such vaccines are expected to address the huge challenges posed by existing conventional vaccines [25]. A recent study has demonstrated that recombinant vaccines have been observed to stimulate long-lived both humoral and cellular immune responses in animals [162]. Vectored vaccines combined the benefits of both replicating and non-replicating subunit vaccines [52,164]. However, the process of developing such a vaccine has been observed to be highly complex and costly [140,163,186].

## 6. Conclusions

The development of prophylactic vaccines and subsequent comprehensive immunization have resulted in partial control of ORFV infection worldwide. However, existing live attenuated vaccines based on traditional attenuation by serial passage of the virus have been found to be inefficient [38,119]. Nevertheless, numerous drawbacks of the existing attenuated ORFV vaccines have been observed, such as low vaccine efficacy, short-lived immunity, and return to virulence [38]. In addition to the poor immunogenicity of current vaccines, there are other critical challenges and knowledge gaps that need to be addressed regarding vaccine safety and efficacy [186,187]. In order to develop safe and effective vaccines against ORFV infections, we envision a new strategy that utilizes recent advances in molecular biology techniques as the basis for developing future vaccines [186]. In recent years, numerous approaches have been used to successfully attenuate the virus through modern genetic manipulation, rather than traditional strategies based on a series of passages of the virus in tissue culture techniques [129,144,186]. As mentioned earlier, the virulence genes of ORFV are known to suppress the host immune response [187]. Therefore, engineering silencing of one or more immunosuppressive genes by genetic modification of the viral genome represents a new strategy for the development of subunits, chimeric DNA [3,27,64] and recombinant vectorized vaccines, the main goal of which is to address issues of safety and efficacy [144,159,188]. Therefore, vaccines will continue to be the basis for new advances in veterinary medicine [65,92,93,94]. We are optimistic that the improvement of current and the development of future vaccines will not only focus on the improvement of viral attenuation and the deletion of immunomodulatory genes, but also on the exploration of immunogenic ORFV proteins that will further enhance the immunogenicity of the new vaccine candidates. If the concept of recombinant vaccines really works, it would be a great strategy to fight ORFV infections [188].

## Figures and Tables

**Figure 1 vaccines-09-01341-f001:**
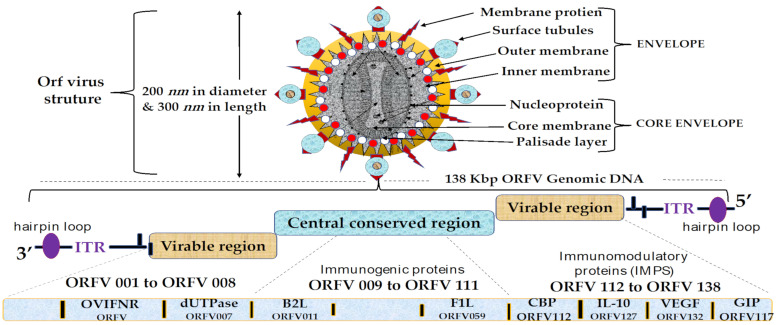
Schematic representation of the organization of ORFV genomic DNA. The 138 kb long linear ds-DNA genome comprises the central conserved region (ORFV009–ORFV111), the right end of the genome represents immunomodulatory proteins (ORFV112 to ORFV138), [84,88] and the left end another variable region (ORFV001 to ORFV008). The nucleocapsid protein holds the viral genome with inverted terminal repeats (ITRs) that closelyassociated with heparin loops at both the 3′ and 5′ ends. The highly variable regions are involved in the virulence and pathogenesis activities of the virus. The conserved region encoding 101 essential genes of the virus involved in immunogenicity, replication, transcription and morphogenesis [109,110,111].

**Figure 2 vaccines-09-01341-f002:**
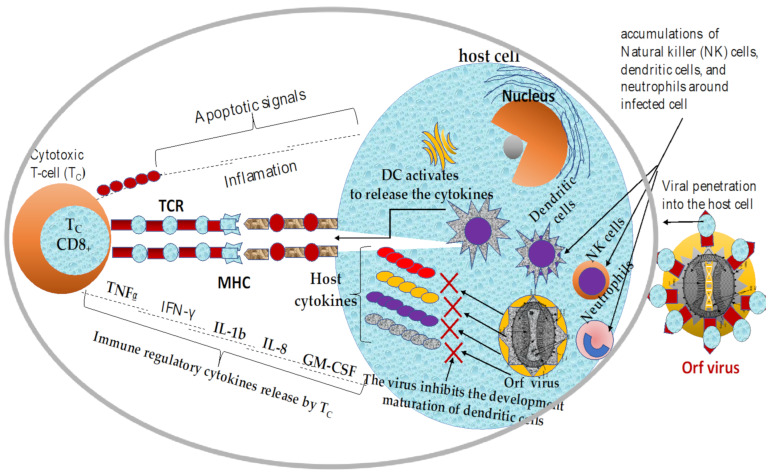
Orf virus invades a susceptible host and temporarily replicates in infected cells by subverting the reactivation of the immune response [102]. Once internalized, innate immune cells such as neutrophils, dendritic cells (DCs), [103] and natural killer cells (NKs) are constantly recruited to sites of infection to engulf the antigens and facilitate migration of the engulfed antigens to peripheral lymph nodes, where antigen presentation to newly recruited naive T cells [105] and memory cells occurs. The DCs present the viral antigen to the specialized cytotoxic CD8^+^ T cells of the immune system, resulting in the proliferation of granulocyte-macrophage colony-stimulating factor (GM-CSF), interleukin-1β (IL-1b), interferon-α (IFN-α), and IL-8 and IFN-γ. The virus can subvert the actions of the host immune response by secreting anti-inflammatory immunomodulatory proteins responsible for neutralizing the virus [106,107,108,122].

**Figure 3 vaccines-09-01341-f003:**
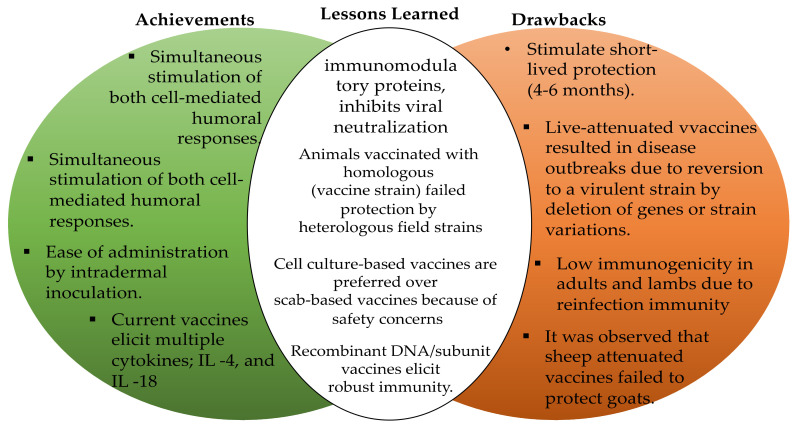
Vein diagram of the overlap of advantages, leasons learned and disadvantages of the immune response elicited by existing vaccine platforms against Orf virus infections.

**Figure 4 vaccines-09-01341-f004:**
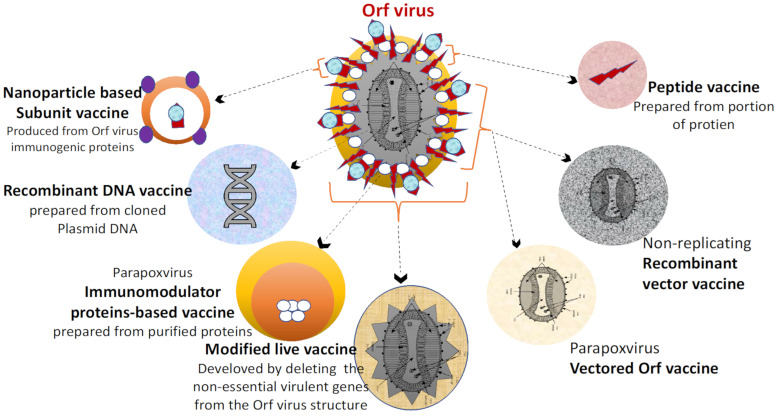
Schematic diagram outlining new strategies to address existing challenges in the design and development of vaccines against viral infections. Modified live attenuated, subunit, recombinant, cytokine and peptide vaccines have been used in modern vaccine development.

**Figure 5 vaccines-09-01341-f005:**
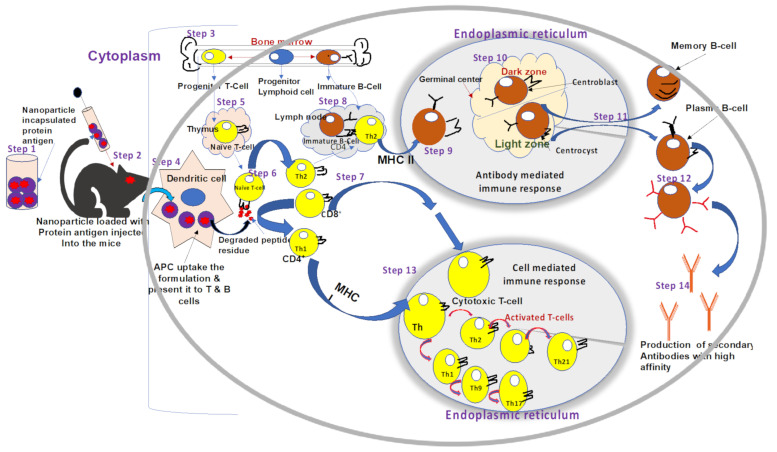
Viral proteins incorporated with nanoparticles of different sizes delivered by immune cells to stimulate T cells and humoral immunity [148,149,150,151,152]. The smaller (10–200 nm) nanoparticle-based vaccines are readily released into the bloodstream and flow on to the lymph nodes for further activation of Th1 and Th2. On the other hand, larger nanoparticles, approximately over 200 nm in size, are first processed by dendritic cells (DCs) and then the antigens can enter the lymph nodes for further presentation and possible activation of immune cells by the small peptide residues released by DCs [153,154]. The small peptide residues bind to the major histocompatibility molecules (MHC) of class I or II, to activate both cellular and humoral immunity responses to foreign challenges.

**Table 1 vaccines-09-01341-t001:** Major immuno-modulator and virulent genes and proteins encoded by ORFV.

Viral	Protein	Gene	Mechanism of Action	Ref.
vVEGF	Vascular endothelial growth factor	ORFV132	inhibit the development and functional maturation of dendritic cells	[95,96,97]
vIL-10	Viral Interleuken-10	ORFV127	inhibits the synthesis and trafficking of host’s cytokines	[98,99,100,106]
vCBP	Viral chemokine binding protein	ORFV112	Stops the cruising and migration of dermal dendritic cells (DCs) to peripheral lymph nodes	[65,87,102,103,104]
GIP	GM-CSF/IL-2 inhibitory factor	ORFV117	inhibits the biological activity of the cytokines GM-CSF and IL-2 (interleukin-2)	[98,101]
OVIFNR	Interferon resistant factor	ORFV020	Inhibits the activities of the cellular IFN	[65,103]

**Table 3 vaccines-09-01341-t003:** Current vaccine platform strategies against Orf virus infections.

Method of Preparation	Target Animal/Year Introduced)	Vaccine Characteristic(s)	Advantages	Disadvantages	Immune Response	Ref.
Cell culture-based vaccine	Sheep/goats (1999,1996,1998, 2008)	attenuated vaccines e.g., D1701 goat vaccine	Induce (4-8 months) humoral & cell-mediated immunity	short-lived immunity, possible reversion to virulence strain	Evokes broad immune responses; and CD4-T cells, CD8-T cells and the cytokine IFN-g (interferon-g) have been observed	[5,14,25,78,129,175]
Egg-based vaccine	Sheep (2008)	Ovine vaccine-egg-based vaccine	Induces antibody mediated immunity	short-lived immunity	ORFV-specific IgG, IFN-γ, IL-4, IL-10, IL-12, and IL-18.	[56,74,78,164]
Scab-based vaccine	sheep (1935, 1989, & 2012)	Homologous goats’ vaccine	4–6-months cell mediated immunity in sheep	potential source of environmental contamination	Cellular immunity plays the main role, IL-4, and IL-18 and O RFV-specific IgG.	[56,120,125,178]
Inactivated Orf vaccine	Sheep/goats	D1701 ORFV	Cell-mediated and humoral immunity	short-lived immunity,	IFN-g & a type 1, IL-4 & IL-10	[59,63,178,183]
Recombinant vaccine	Sheep/goats (2011, 2016, & 2019)	Chimeric DNA vaccine	Mimics adaptive immunity	required multiple boosters	IFN-γ, IL-4, IL-18 and ORFV-specific IgG.	[25,160,130,183]

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
