# Peer review of "Immunomodulatory Strategies for Parapoxvirus: Current Status and Future Approaches for the Development of Vaccines against Orf Virus Infection"

_vaccines, 2021, doi:10.3390/vaccines9111341_

Round 1
Reviewer 1 Report
The authors made a thorough revision to their manuscript and have adequately addressed my previous concerns.
Author Response
Response to Reviewer 1 Comments
The authors made a thorough revision to their manuscript and have adequately addressed my previous concerns.
[Authors' Reply]: We greatly appreciate your highly positive comments.

Reviewer 2 Report
The manuscript was greatly enhanced by the authors. Now the manuscript is easier to read and complete.
There are minor details in the text to be corrected; however, those corrections may be done when the manuscript is being edited.
Author Response
Response to Reviewer 2 Comments
The manuscript was greatly enhanced by the authors. Now the manuscript is easier to read and complete.
[Authors' Reply]: We greatly appreciate your highly positive comments.
There are minor details in the text to be corrected; however, those corrections may be done when the manuscript is being edited.
[Reply]: We greatly appreciate your comments. We apologize for our error, which has been improved in the revised manuscript. All additions in the manuscript have been marked with blue color for your approval.

This manuscript is a resubmission of an earlier submission. The following is a list of the peer review reports and author responses from that submission.
Round 1
Reviewer 1 Report
The review entitled immunomodulatory Strategies for Parapoxvirus Current
Status and Future Approaches for Development of Efficacious Vaccines against Orf Virus infection, in general, represents an ideal important issue to discuss in the immunology field and in the veterinary field. However, the review lacks a good structure and the title does not reflect the content. There are several redundancies in the descriptions and the immunomodulatory response are described and not well analyzed. The structure of the review should consider first the structure and the analysis of the virus itself; then, immunogenic proteins, innate response, adaptative cellular response. Point 2 describes the immune response against the virus and then point 2.1 the effect of viral proteins in which critical information is missing. Figure 1 also has to be corrected since it depicts a host cell and within that cell other host cells. The amount of critical inflammatory cytokines is scarce as well as there is no information on antigen presentation. Then, the authors describe the different vaccines used and a general analysis of immune response to the vaccines in table 2 and figure 2 which is scarce and should be enhanced. Figures 3 and 4 can be deleted since it does not provide important information. The relevant information is in table 3 concerning the T cell immune response, there is no discussion by the authors about why the vaccines are not efficient. A general, not well-argued genetic variation of the virus follows without even discussing how the current vaccines could be enhanced. Finally the last strategies a just merely discussed without the proper immunological analysis. It can be concluded then that the title of the review and the text do not match.
There are several minor issues, the abstract is long and it does not provide the critical information required and the conclusions have to be rewritten. In general the whole manuscript has to be written based on the suggestions
Reviewer 2 Report
In the manuscript by Bukar et al., the authors thoroughly review vaccine strategies against ORFV. They describe many current methods of vaccination against any type of virus, then describe current and future strategies against ORFV. Overall, the manuscript is very thorough, but sometimes not very focused on ORFV. Some of the figures are unrelated to ORFV and redundant. Below I outline the points the authors need to address in their revision.
-Abstract: the authors should briefly describe Orf disease. In general, the abstract is too long and unfocused. The major points of the manuscript should be summarized here more concisely.
-Line 121: are the low CD4+ and CD8+ T-cells specific for ORFV antigens or do the authors mean levels of these cells in general?
-Line 143: Is this a reference to Figure 1? If so, I do not see how this figure is related to B2L, as stated in line 142.
-Lines 157 and 159: what are IMPs and IMGs? These may be defined later on, but they need to be defined at the first acronym use.
-Tables 1-3 are not called out or referenced anywhere in the main text.
-Section 3: At the end of each subsection, it would be helpful to the reader if the authors state if any of the vaccine strategies described are used against ORFV.
-Table 2: Peptide and recombinant vaccines are not described in Section 3, but rather section 6, so showing these in the table before they are described is confusing.
-Section 4: how effective are the current vaccine strategies against ORFV? Knowing these statistics would be very beneficial to the reader.
-Figures 2, 3, 5 are not called out or referenced in any part of the main text.
-Figure 4: The purpose of including this general figure about vaccination history is not clear. It should be removed.
-Lines 308-315: Is this section specific to ORFV or viruses in general?
-Table 3 and all discussion about contagious ecthyma: It is not clear why CE is included in this review. Is this caused by a virus? Everything about CE should be removed from the manuscript.
Overall: The manuscript needs a major revision so that it is more focused on ORFV, as stated in the title. Otherwise, the manuscript reads like a review of vaccines in general with a slight focus on ORFV. The manuscript also needs major editing for grammatical errors.